# GS-Bias: Global-Spatial Bias Learner for Single-Image Test-Time Adaptation of Vision-Language Models

Zhaohong Huang [1]   Yuxin Zhang [1]   Jingjing Xie [1]   Fei Chao [1]   Rongrong Ji [1]

## Abstract

Recent advances in test-time adaptation (TTA) for Vision-Language Models (VLMs) have garnered increasing attention, particularly through the use of multiple augmented views of a single image to boost zero-shot generalization. Unfortunately, existing methods fail to strike a satisfactory balance between performance and efficiency, either due to excessive overhead of tuning text prompts or unstable benefits from handcrafted, training-free visual feature enhancement. In this paper, we present Global-Spatial Bias Learner (GS-Bias), an efficient and effective TTA paradigm that incorporates two learnable biases during TTA, unfolded as the global bias and spatial bias. Particularly, the global bias captures the global semantic features of a test image by learning consistency across augmented views, while the spatial bias learns the semantic coherence between regions in the image's spatial visual representation. It is worth highlighting that these two sets of biases are directly added to the logits output by the pretrained VLMs, which circumvent the full backpropagation through VLM that hinders the efficiency of existing TTA methods. This endows GS-Bias with extremely high efficiency while achieving state-of-the-art performance on 15 benchmark datasets. For example, it achieves a 2.23% improvement over TPT in cross-dataset generalization and a 2.72% improvement in domain generalization, while requiring only 6.5% of TPT's memory usage on ImageNet. Our code is released at https://github.com/hzhxmu/GS-Bias.

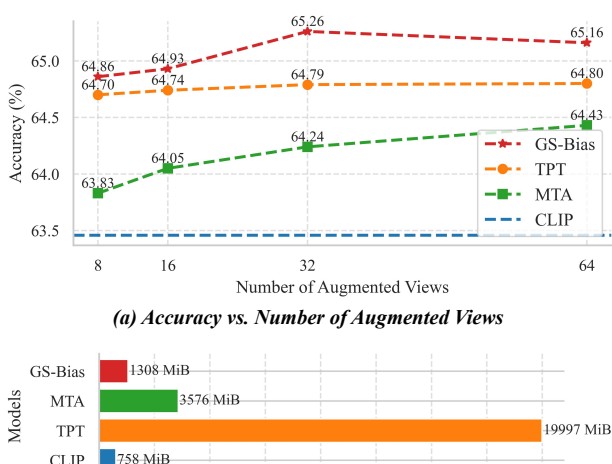

*(a) Accuracy vs. Number of Augmented Views*

*(b) GPU Memory Usage Comparison*

*Figure 1.* (a) Comparison of state-of-the-art TTA methods (Shu et al., 2022; Zanella & Ben Ayed, 2024) and our proposed GS-Bias on 10 cross-datasets generalization benchmarks (Maji et al., 2013; Fei-Fei et al., 2004; Krause et al., 2013; Cimpoi et al., 2014; Nilsback & Zisserman, 2008; Bossard et al., 2014; Helber et al., 2019; Sun et al., 2020; Soomro, 2012; Parkhi et al., 2012). (b) Comparison of memory consumption of different TTA paradigms on the ImageNet (Deng et al., 2009), where all methods are evaluated using 64 augmented views of the test sample.

## 1. Introduction

In recent years, the development of large-scale vision-language pre-trained models (VLMs) has significantly accelerated progress within the computer vision community (Jia et al., 2021; Alayrac et al., 2022; Radford et al., 2021b). Notably, CLIP (Radford et al., 2021a), trained on 0.4 billion text-image pairs, have demonstrated remarkable zero-shot performance across a wide range of downstream tasks. This success has ignited further research interest in the transfer learning capabilities of VLMs. Prevailing efforts mainly falls on tuning VLM prompt (Zhou et al., 2022b; Chen et al., 2023; Fu et al., 2024) or using visual adapters (Gao et al., 2024; Zhang et al., 2022b; Udandarao et al., 2023) to enhances the efficacy of VLMs on downstream tasks. However, two significant limitations are apparent (Zhang et al., 2022a; Wang et al., 2021; Chen et al., 2024; Lin

---

[1]Key Laboratory of Multimedia Trusted Perception and Efficient Computing, Ministry of Education of China, Xiamen University, 361005, P.R. China. Correspondence to: Rongrong Ji <rrji@xmu.edu.cn>.

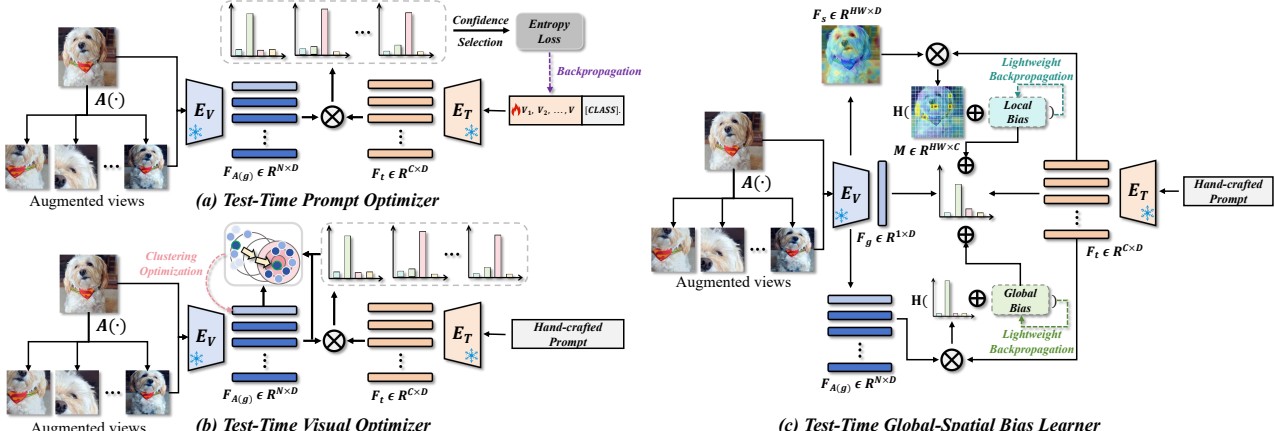

Figure 2. Illustration of (a) Test-Time Prompt Optimizer, (b) Test-Time Visual Optimizer and (c) our proposed Test-Time Global-Spatial Bias Learner (GS-Bias). GS-Bias introduces two sets of learnable, independent global and spatial biases directly on the prediction distribution of a single image, *i.e.*, the output of CLIP. By optimizing both biases solely at the logits output stage, our GS-Bias achieves lightweight backward operations and eliminates the overhead of secondary inference.

et al., 2025): (1) the zero-shot generalization capabilities of VLMs are severely compromised due to the substantial distribution shifts between the training data and downstream tasks, and (2) these approaches typically depend on access to high-quality, task-specific training data, which is often impractical in real-world scenarios.

In response to these drawbacks, recent works have shifted towards a new paradigm known as test-time adaptation (TTA) (Shu et al., 2022; Feng et al., 2023; Zanella & Ben Ayed, 2024; Yoon et al., 2024; Liu et al., 2024). TTA addresses the above obstacles of domain shift and training data dependency by enabling VLMs to dynamically align with individual test samples on-the-fly. To achieve this, existing methods mainly leverage multiple augmented views of a single test image and optimize either text prompts or image features for the sample-wise adaption, which we respectively discussed as follows.

**Test-Time Prompt Optimizer**, as depicted in Figure 2 (a), optimizes the VLM textual prompts by minimizing the marginal entropy of augmented views derived from the test sample (Yoon et al., 2024; Shu et al., 2022; Feng et al., 2023; Liu et al., 2024). Despite its promising performance, modifying textual prompts can lead to massive memory and computational overhead (see Figure 1 (b)). This is due to the intensive backward propagation operations across the full network during the CLIP input-level optimization. Furthermore, the updated prompts require a second forward pass for inference, which is time-consuming (Zanella & Ben Ayed, 2024).

**Test-Time Visual Optimizer**, as shown in Figure 2 (b), focuses on optimizing the internal visual features of VLMs. For example, MTA (Zanella & Ben Ayed, 2024) utilizes

MeanShift (Comaniciu & Meer, 1999) to jointly optimize visual features by leveraging internal scores and density modes across multiple augmented views. Compared to prompt learning, such visual optimizer demonstrated higher efficiency due to its training-free nature. However, its performance strongly depends on the quality of augmented views generated by the pre-trained model, as demonstrated in Figure 1 (a), which limits its efficacy in zero-shot, complex real-world scenarios, such as those involving unseen classes.

To sum up, the two paradigms discussed above still fail to achieve an optimal trade-off between efficiency and performance. Figure 1 compares performance and efficiency of representative methods TPT (Shu et al., 2022) and MTA (Zanella & Ben Ayed, 2024) for prompt and visual optimizers, respectively. Prompt optimizer exhibits superior performance by learning new knowledge specific to individual images. In contrast, visual optimizer provides higher efficiency through parameter-free optimization of internal features, albeit with weaker performance constrained by the quality of the augmented images.

In response to the above drawbacks, this paper presents Global-Spatial Bias Learner (GS-Bias), a novel TTA method that equips learnable biases directly alongside the output logits of VLMs. Particularly, GS-Bias contains two types of biases that plays unique role in improving the TTA performance, unfolded as the global and local bias. The global bias captures the global semantic knowledge of the test image by enforcing the consistency of logits across different views. For the local bias, we leverage the spatial visual representations from the visual encoder to provide contextual information (Miyai et al., 2024; Lafon et al., 2024). Specifically, we select regions highly relevant to the downstream

task for spatial semantic consistency learning, which is used to update the spatial bias. As illustrated in Figure 2 (c), the cost of updating the biases is extremely low, as the backward propagation occurs only at the output side, eliminating the need to further propagate through the entire network as previous works do (Shu et al., 2022; Feng et al., 2023; Yoon et al., 2024; Liu et al., 2024). Consequently, as shown in Figure 1, GS-Bias achieves a well-balanced trade-off between performance and efficiency of TTA. For example, with settings aligned to TPT (*i.e.*, using the prompt `"a photo of a [class]."` and 64 augmented views), GS-Bias achieves leading performance across 10 cross-dataset benchmarks, while consuming merely 6.5% of TPT's memory overhead on ImageNet. Our contributions in this paper include:

- We propose Global-Spatial Bias Learner (GS-Bias), which unifies global semantics across augmented views via global bias and aligns regional semantics within spatial visual representations through spatial bias.

- GS-Bias innovatively performs optimization at the logit level of network outputs, enabling lightweight inference and drastically reducing the TTA overhead of existing methods.

- We report comprehensive evaluations and comparisons to the existing TTA method for VLMs across 15 datasets, demonstrating that GS-Bias achieves an excellent balance between performance and efficiency.

## 2. Related works

### 2.1. Contrastive Vision-Language Models

The core of contrastive visual-language models (Radford et al., 2021a; Jia et al., 2021; Radford et al., 2021b) lies in mapping text and images into a unified space, thereby enabling a more comprehensive understanding of visual concepts. Among these models, CLIP (Radford et al., 2021a) distinguishes itself through its strong zero-shot capabilities across a wide range of downstream tasks. This success has spurred extensive interest in transferring its pretrained knowledge to more complex tasks. A primary area of focus has been devoted to prompt tuning for few-shot learning scenarios (Zhou et al., 2022b;a; Chen et al., 2023; Lafon et al., 2024; Miyai et al., 2024; Fu et al., 2024). For instance, CoOp (Zhou et al., 2022b) replaces handcrafted prompts with learnable context tokens for task adaptation, and CoCoOp (Zhou et al., 2022a) further incorporates image-specific conditional information into the optimization vectors. Differently, adapter-based methods (Gao et al., 2024; Zhang et al., 2022b; Udandarao et al., 2023) focus on designing extra lightweight modules to adapt VLMs. For example, CLIP-Adapter (Gao et al., 2024) introduces learn-

able layers in both branches to modify feature distribution, while Tip-Adapter (Zhang et al., 2022b) uses memory mechanism to perform training-free few-shot setups. Despite their advantageous performance enhancements, these methods necessitate considerable computational resources and training data, which hinders their practical deployment in real-world contexts. In this paper, we investigate test-time adaptation in VLMs, which does not require prior access to downstream training data.

### 2.2. Test-Time Adaptation for VLMs

In real-world scenarios, the inputs to VLMs are often heterogeneous, underscoring the critical need for test-time adaptation (TTA), particularly in contexts such as autonomous driving under adverse weather conditions (Lin et al., 2025) or the diagnosis of diseases during their progression (Chen et al., 2024). The most prevalent mechanism for achieving TTA involve minimizing entropy across either across a batch of test samples (Wang et al., 2021) or across multiple augmented views of a single test sample (Zhang et al., 2022a). As a pioneering work, TPT (Shu et al., 2022) fine-tunes the textual context prompts based on the above objective. DiffTPT (Feng et al., 2023) further introduces diffusion model to diversify the augmentation pool of the test images. Despite the efficacy, such prompt tuning can lead to massive memory usage and computation resources due to the backward propagation operation through the entire network and secondary inference. In response, MTA (Zanella & Ben Ayed, 2024) transitions TTA from soft prompt learning to direct visual optimization by refining the visual features through internal evaluations of augmented views, thus demonstrating greater adaptation efficiency. However, existing TTA methods still fail to achieve favorable trade-off between computational efficiency and performance generalization. In this work, we propose GS-Bias, which introduces lightweight global and spatial biases. These biases capture global semantics through augmented view consistency and align spatial information via regional consistency in visual representations. Crucially, the biases are optimized solely at the logit level, eliminating full-network backward propagation and redundant secondary inference, which significantly reduces computational overhead while enhancing generalization performance.

## 3. Method

### 3.1. Background

**Vision Language Models.** By aligning image and text within a unified embedding space, Vision-Language Models (VLMs) such as CLIP (Radford et al., 2021b) mark substantial advancements across a broad range of vision-related tasks. Without loss of generality, We utilize CLIP, equipped with a Vision Transformer (ViT (Dosovitskiy, 2020)), as the

foundational model for our methodological demonstration. CLIP is composed of a visual encoder $\mathbf{E_V}$ and a text encoder $\mathbf{E_T}$, which independently extract information from images and text, subsequently mapping them into a shared feature space. During training, a contrastive loss (Chen et al., 2020) is employed to maximize the cosine similarity between matched pairs from different modalities.

For a specific downstream task involving $C$ classes, each class is embedded into a hard text prompt to generate the corresponding text embeddings $\{\boldsymbol{F}_t^c\}_{c=1}^C$, where $\boldsymbol{F}_t^c \in \mathbb{R}^d$ denotes the text feature of the class-specific text input. At test time, given a single input image $\boldsymbol{x}$, the visual encoder extracts a global visual representation $\boldsymbol{F}_g = \mathbf{E_V}(\boldsymbol{x}) \in \mathbb{R}^d$, where $d$ is the feature dimension. By computing the cosine similarity between the visual feature and text embeddings, the prediction probabilities for the query image $\boldsymbol{x}$ with respect to class $y_c$ can be derived as:

$$p_{\text{CLIP}}(y_c|\boldsymbol{x}) = \frac{\exp\left(\cos\left(\boldsymbol{F}_g, \boldsymbol{F}_t^c\right)/\tau\right)}{\sum_{c=1}^C \exp\left(\cos\left(\boldsymbol{F}_g, \boldsymbol{F}_t^c\right)/\tau\right)}, \quad (1)$$

where $cos(\cdot)$ calculates the cosine similarity between vectors, and $\tau$ is the temperature of the softmax function.

**Revisiting Test-time Adaption of VLMs.** CLIP's output can be further refined through tuning text prompts or visual adapters based on specific downstream datasets (Zhou et al., 2022b;a; Zhang et al., 2022b; Gao et al., 2024). However, these approaches require substantial training costs and data overhead, thus rendering it impractical in real-world scenarios. As a more efficient way, test-time adaption adjusts the output of CLIP on-the-fly. This is typically achieved by using multiple augmented views of the test image, denoted as $\{\mathcal{A}_n(\boldsymbol{x})\}_{n=1}^N$, where $\mathcal{A}(\cdot)$ is the augmentation function, and $N$ denotes the number of augmented views, to adapt the model for more convincing output. For a representative example, TPT (Shu et al., 2022) tunes the text prompt $\boldsymbol{V}$ to reduce inconsistencies in the model's predictions across these diverse augmentations. Particularly, the augmented views are firstly leveraged to generate a set of output logits as:

$$\tilde{p}\left(y_c|\boldsymbol{x}\right) = \frac{1}{\rho N} \sum_{i=1}^N \mathbb{I}\left[\mathbf{H}\left(p_i\right) \leq \theta\right]\left(p_i\left(y_c \mid \mathcal{A}_i(\boldsymbol{x})\right)\right), \quad (2)$$

where $\mathbf{H}(\cdot)$ calculates the self-entropy of the predicted probability distribution, and the indicator function $\mathbb{I}\left[\mathbf{H}\left(p_i\right) \leq \theta\right]$ filters high-uncertainty logits smaller than $\theta$. Afterwards, TPT updates the text prompt $\boldsymbol{V}$ by minimizing the distribution entropy of top-$\rho$ confident samples $\tilde{p}\left(y_c|\boldsymbol{x}\right)$ as:

$$\mathbf{H}\left(\tilde{p}\right) = -\sum_{c=1}^C \tilde{p}_g\left(y_c \mid \boldsymbol{x}\right) \log \tilde{p}_g\left(y_c \mid \boldsymbol{x}\right), \quad (3)$$

$$\boldsymbol{V}^* \leftarrow \boldsymbol{V} - \eta \nabla_{\boldsymbol{V}} \mathbf{H}\left(\tilde{p}\right), \quad (4)$$

where $\eta$ represents the learning rate for gradient descent, while $\nabla_{\boldsymbol{V}} \mathbf{H}\left(\tilde{p}\right)$ denotes the gradient of the entropy. The test-specific prompt $\boldsymbol{V}^*$ is then utilized as the text prompt for CLIP during secondary inference, generating the corresponding test-time text embeddings $\{\boldsymbol{F}_{\boldsymbol{V}^*}^c\}_{c=1}^C$. Consequently, the final predictions of CLIP is refined as:

$$p_{\text{TPT}}(y_c|\boldsymbol{x}) = \frac{\exp\left(\cos\left(\boldsymbol{F}_g, \boldsymbol{F}_{\boldsymbol{V}^*}^c\right)/\tau\right)}{\sum_{c=1}^C \exp\left(\cos\left(\boldsymbol{F}_g, \boldsymbol{F}_{\boldsymbol{V}^*}^c\right)/\tau\right)}. \quad (5)$$

Despite the promising performance of such test-time prompt optimizers, learning the prompts requires backpropagating gradients through the entire network and involves redundant secondary inferences, posing significant challenges to computational efficiency. Recently proposed visual optimizers (Zanella & Ben Ayed, 2024) offer lightweight solutions by directly integrating the augmented views using clustering methods like Mean-shift (Comaniciu & Meer, 1999) to boost the robustness of visual embeddings. Unfortunately, the performance of such parameter-free optimization drastically degrade when applied to domains outside of CLIP's pretraining knowledge, such as unseen classes.

### 3.2. Global-Spatial Bias Learner

In this work, we propose Global-Spatial Bias Learner (GS-Bias) as a way to address the above trade-off issue in performance and efficiency. The core contribution of GS-Bias lies in introducing two learnable bias terms $\boldsymbol{B}_g$ and $\boldsymbol{B}_s$ during VLM test-time adaption. As illustrated in Fig. 1 (c), the global bias $\boldsymbol{B}_g$ enhances global semantic knowledge by adjusting the logit outputs of each augmented view, whereas the local bias $\boldsymbol{B}_s$ refines the local spatial knowledge by selectively emphasizing the visual patches most relevant to class-specific objects. Notably, both $\boldsymbol{B}_g$ and $\boldsymbol{B}_s$ are optimized at the logits output stage, thus eliminating the need for entire network backpropagation and secondary inference. We detail the mechanism and implementation of the global and local bias terms of GS-Bias as below.

**Global Bias Learner.** As previously discussed, test-time prompt optimizers tunes text prompt to refine the logits entropy of augmented views (Shu et al., 2022). Although such global enhancement of semantic knowledge can lead to performance improvements, optimizing the text prompts at the model's input side clearly lead to efficiency bottleneck. We address this obstacle by directly adding a lightweight global bias learner to the model's output side. Specifically, to capture global bias knowledge within a single image, we begin by zero-initializing a learnable bias term $\boldsymbol{B}_g \in \mathbb{R}^{1 \times C}$, which is shaped to match the distribution of CLIP's final output. The learnable global bias is shared across all augmented views and added to their respective logits, acting as a global logit to capture the image's global semantic

information, written as:

$$\tilde{p}_g\left(y_c|\boldsymbol{x}\right) = \frac{1}{\rho N}\sum_{i=1}^{N}\mathbb{I}\left[\mathbf{H}\left(p_i\right) \leq \theta\right]\left(p_i\left(y_c \mid \mathcal{A}_i(\boldsymbol{x})\right) + \boldsymbol{B}_g\right). \tag{6}$$

Then, we follow TPT to calculates the self-entropy of the predicted probability distribution $\mathbf{H}\left(\tilde{p}_g\right)$ as per Eq. 2, while perform very lightweight backpropagation to update the learnable global bias as:

$$\boldsymbol{B}_g \leftarrow \boldsymbol{B}_g - \alpha\nabla_{\boldsymbol{B}_g}\mathbf{H}\left(\tilde{p}_g\right), \tag{7}$$

where $\alpha > 0$ controls the strength of the global consistency constraint. After updating, we directly add the global bias to the vanilla output logit as:

$$p_{\mathrm{G}}(y|\boldsymbol{x}) = p_{\mathrm{CLIP}}(y|\boldsymbol{x}) + \boldsymbol{B}_g. \tag{8}$$

It is worth noting that the updating and incorporation of $\boldsymbol{B}_g$ occur directly at the output end of the text branch, thus eliminating the need for complete backpropagation and subsequent forward propagation post-update.

**Spatial Bias Learner.** We further present spatial bias learner to enhance to local spatial feature within the image branch of VLMs. In particular, spatial features $\boldsymbol{F}_s \in \mathbb{R}^{wh \times d}$ refer to the intrinsic embeddings within the vision encoder, where $h = \frac{H}{P}$ and $w = \frac{W}{P}$, with $H$ and $W$ representing the height and width of the input image, respectively, and $P$ denoting the patch size. Recent advancements in prompt tuning (Miyai et al., 2024; Lafon et al., 2024) have shown that enriching such spatial features brings significant performance gains. Despite this progress, the potential of spatial features remains untapped in TTA settings. To address this gap, we introduce a set of zero-initialized learnable spatial biases $\boldsymbol{B}_s \in \mathbb{R}^{1 \times C}$ to capture spatial knowledge specific to a single image. Since the visual encoder maps both spatial features $\boldsymbol{F}_s$ and text embeddings $\boldsymbol{F}_t$ to a feature space with the same dimensionality, we can directly compute the prediction probability for each spatial feature region with respect to class $y_c$, resulting in a set of spatial probabilities $\boldsymbol{S} = \{\boldsymbol{p}(y_c \mid \boldsymbol{F}_s^i)\}_{i=1}^{w \times h}$, defined as:

$$p(y_c \mid \boldsymbol{F}_s^i) = \frac{\exp\left(\cos\left(\boldsymbol{F}_s^i, \boldsymbol{F}_t^c\right)/\tau\right)}{\sum_{c=1}^{C}\exp\left(\cos\left(\boldsymbol{F}_s^i, \boldsymbol{F}_t^c\right)/\tau\right)}. \tag{9}$$

Subsequently, we identify which spatial regions of the test sample are most relevant to the target classes and designate these regions as spatial views for further optimization. Specifically, we compute a category-aware spatial feature map $\boldsymbol{M}$ as

$$\boldsymbol{M} = \frac{1}{C}\sum_{c=1}^{C}\mathrm{Softmax}\left(\boldsymbol{F}_s(\boldsymbol{F}_t^c)^T\right), \tag{10}$$

$\boldsymbol{M} \in \mathbb{R}^{wh \times 1}$ represents a set of spatial sequences enriched with category information, where each element indicates

the concentration of relevance between the corresponding region and the classes in the downstream task. Intuitively, elements with higher concentration values are likely to be more significant. Thus, we employ a straightforward Top-K selection strategy to filter out the most informative regions:

$$\boldsymbol{I} = \{i \mid \mathrm{Top}\text{-}K(\boldsymbol{M}_i), i \in \{1, 2, \ldots, w \times h\}\}, \tag{11}$$

$\boldsymbol{I}$ denotes the set of indices corresponding to the selected spatial regions. Using these indices, we retrieve the selected probabilities from the regions-text probability set $\boldsymbol{S}$, thereby obtaining a refined set of predictions for the most relevant regions $\tilde{\boldsymbol{S}} = \{p(y_c \mid \boldsymbol{F}_s^i)\}_{i=1}^{i \in \boldsymbol{I}}$. The set $\tilde{\boldsymbol{S}}$ contains the top $K$ regions most likely to contain the class-specific object. For clarity, we denote this collection as $\hat{\boldsymbol{S}} = \{p(y_c \mid \boldsymbol{F}_s^k)\}_{k=1}^{K}$. We strategically incorporate a shared spatial bias $\boldsymbol{B}_s$ into $\hat{\boldsymbol{S}}$, thereby generating spatial logits $\tilde{p}_s(y_c \mid x)$ that serve as a foundation for exploring the fine-grained spatial knowledge within the test image:

$$\tilde{p}_s\left(y_c|\boldsymbol{x}\right) = \frac{1}{K}\sum_{k=1}^{K}\left(p_k\left(y_c \mid \boldsymbol{F}_s^k\right) + \boldsymbol{B}_s\right). \tag{12}$$

Then, we enforce spatial consistency by calculating the entropy of $\tilde{p}_s$ in the spatial branch as:

$$\mathbf{H}\left(\tilde{p}_s\right) = -\sum_{c=1}^{C}\tilde{p}_s\left(y_c \mid \boldsymbol{x}\right)\log\tilde{p}_s\left(y_c \mid \boldsymbol{x}\right). \tag{13}$$

The underlying principle of the above learning target is that the selected high-information regions should encapsulate the majority of the target class information. At last, we update the spatial bias by computing the gradient of Eq. 9 as:

$$\boldsymbol{B}_s \leftarrow \boldsymbol{B}_s - \beta\nabla_{\boldsymbol{B}_s}\mathbf{H}\left(\tilde{p}_s\right), \tag{14}$$

where $\beta$ represents the learning rate of the spatial bias. Based on the above, we have obtained the global bias $\boldsymbol{B}_g$ summarizing the overall semantics, the spatial bias $\boldsymbol{B}_s$ containing fine-grained details of the test image, and the pretrained CLIP output distribution $p_{\mathrm{CLIP}}(y|\boldsymbol{x})$. These components are combined to produce the final output of GS-Bias during the test phase, as follows:

$$p_{\mathrm{GS}\text{-}\mathrm{Bias}}(y|\boldsymbol{x}) = p_{\mathrm{CLIP}}(y|\boldsymbol{x}) + \boldsymbol{B}_g + \boldsymbol{B}_s. \tag{15}$$

## 4. Experiment

### 4.1. Experimental Setup

**Benchmarks.** Following standard practices in Test-Time Adaptation (TTA) (Shu et al., 2022; Feng et al., 2023; Zanella & Ben Ayed, 2024), our primary results encompass two key benchmarks for model generalization: Cross-Datasets Generalization and Domain Generalization. For

*Table 1.* Comparison of GS-Bias in Cross-Datasets Generalization using ViT-B/16 (Dosovitskiy, 2020) as the backbone. Bold values indicate the best performance, while † and ‡ refer to results reported in (Shu et al., 2022) and (Feng et al., 2023), respectively. E. denotes the use of hand-crafted prompts suggested in CLIP (Radford et al., 2021a). $BS$ represents the number of augmented views.

| Method | Flowers102 | DTD | Pets | Cars | UCF101 | Caltech101 | Food101 | SUN397 | Aircraft | EuroSAT | Average |
|---|---|---|---|---|---|---|---|---|---|---|---|
| CLIP-ViT-B/16 | 67.28 | 44.44 | 88.06 | 65.28 | 65.03 | 92.94 | 83.82 | 62.59 | 23.82 | 41.38 | 63.46 |
| CLIP + E. | 71.34 | 44.39 | 89.10 | 65.91 | 66.67 | 94.12 | 86.01 | 66.29 | 24.84 | 47.72 | 65.64 |
| CoOp† | 68.71 | 41.92 | 89.14 | 64.51 | 66.55 | 93.70 | 85.30 | 64.15 | 18.47 | 46.39 | 63.88 |
| CoCoOp† | 70.85 | 45.45 | **90.46** | 64.90 | **68.44** | 93.79 | 83.97 | 66.89 | 22.29 | 39.23 | 64.63 |
| TPT ($BS = 64$) | 69.31 | 46.99 | 87.38 | 65.99 | 68.01 | 94.10 | 84.73 | 65.43 | 23.27 | 42.81 | 64.80 |
| DiffTPT‡ ($BS = 128$) | 70.10 | **47.00** | 88.22 | 67.01 | 66.69 | 92.49 | **87.23** | 65.74 | 25.60 | 43.13 | 65.47 |
| MTA ($BS = 64$) | 67.64 | 45.15 | 87.90 | 67.31 | 68.22 | 94.00 | 84.61 | 65.19 | 23.91 | 41.35 | 64.43 |
| MTA + E. ($BS = 64$) | 71.34 | 44.68 | 89.37 | 66.43 | 67.33 | 94.32 | 86.25 | 66.82 | 25.23 | 47.75 | 65.95 |
| GS-Bias ($BS = 8$) | 68.86 | 45.10 | 88.58 | 66.77 | 65.74 | 94.16 | 85.67 | 64.78 | 25.30 | 43.63 | 64.86 |
| GS-Bias + E. ($BS = 8$) | **71.94** | 46.10 | 90.38 | **67.33** | 67.59 | **94.60** | 86.09 | **67.40** | **26.49** | **52.42** | **67.03** |

the Cross-Datasets Generalization benchmark, we assess performance on 10 diverse classification datasets, covering a broad spectrum of visual recognition tasks. These include datasets for plant and animal species (Flowers102 (Nilsback & Zisserman, 2008) and OxfordPets (Parkhi et al., 2012)), transportation (StanfordCars (Krause et al., 2013) and FGVC-Aircraft (Maji et al., 2013)), food (Food101 (Bossard et al., 2014)), satellite imagery (EuroSAT (Helber et al., 2019)), human actions (UCF101 (Soomro, 2012)), texture (DTD (Cimpoi et al., 2014)), scene recognition (SUN397 (Sun et al., 2020)), and general object classification (Caltech101 (Fei-Fei et al., 2004)). In the Domain Generalization setting, we evaluate our approach on four out-of-distribution (OOD) variants of ImageNet (Deng et al., 2009): ImageNetV2 (Recht et al., 2019), ImageNet-Sketch (Wang et al., 2019), ImageNet-A (Hendrycks et al., 2021b), and ImageNet-R (Hendrycks et al., 2021a).

**Baselines.** We compare our proposed method, GS-Bias, with several state-of-the-art approaches across two benchmarks. The comparison includes zero-shot CLIP (Radford et al., 2021a), two representative training-time adaptation methods (CoOp (Zhou et al., 2022b) and CoCoOp (Zhou et al., 2022a)), and three test-time adaptation methods (TPT (Shu et al., 2022), DiffTPT (Feng et al., 2023), and MTA (Zanella & Ben Ayed, 2024)), all tailored for vision-language models. Notably, CoOp and CoCoOp are trained on the ImageNet (Deng et al., 2009) dataset, with 16 views per class, and then transferred to downstream datasets for evaluation. In contrast, test-time adaptation methods perform adaptation for each test sample without the need to access the training data.

**Implementation details.** In all experiments, we use the publicly available pre-trained CLIP model, with ViT-B/16 (Dosovitskiy, 2020) as the backbone and a Transformer-based text encoder (Vaswani, 2017). For each test image, we evaluate two hand-crafted prompts: the basic prompt `"a photo of a [class]."` and the more

elaborate ensemble prompt described in (Radford et al., 2021a). It is important to highlight that due to the learnable nature of textual descriptions in the prompt optimizer, implementing an ensemble setup in this context is not straightforward. Therefore, we incorporate the ensemble of prompts separately into our approach. Aligned with prior TTA works (Shu et al., 2022; Zanella & Ben Ayed, 2024), we adopt random cropping as the data augmentation strategy. Specifically, for the unseen cross-dataset generalization, we obtain a set of augmented views with $BS = 8$ and set the views selection rate to $\rho = 0.5$. For the out-of-distribution domain generalization, the number of augmented views is increased to $BS = 64$, and the views selection rate is adjusted to $\rho = 0.3$. For the learning of GS-Bias, the number of important spatial regions $K$ in Eq.12 is fixed at 16. These biases are optimized with 5 steps during test phase. The learning rates for the biases in Eq. 7 and Eq. 14 are set to $\alpha = 1$ and $\beta = 1$ for cross-domain generalization, whereas for domain generalization, $\alpha = 10$ and $\beta = 1$. Across all experiments, top-1 accuracy (%) is used as the evaluation metric, which is a standard measure for classification performance.

### 4.2. Comparisons with State-of-the-art

**Results on the Cross-Datasets Generalization.** We report the quantitative results of various methods for cross dataset generalization on 10 benchmarks. Table 1 shows our GS-Bias achieves the highest average accuracy and demonstrates superior performance in 6 out of the 10 datasets. In particular, GS-Bias consistently outperforms CLIP across all 10 datasets, a performance trend not observed with other baselines. For example, TPT experiences a performance drop on the Aircraft dataset, while MTA faces a similar issue on the Pets dataset. This highlights the potential risks of modifying the intrinsic visual and textual representations of pre-trained CLIP, which may lead to performance degradation. In contrast, our method enhances performance by

*Table 2.* Comparison of GS-Bias in Domain Generalization using ViT-B/16 (Dosovitskiy, 2020) as the backbone. Bold values indicate the best performance, while † and ‡ refer to results reported in (Shu et al., 2022) and (Feng et al., 2023), respectively. E. denotes the use of hand-crafted prompts suggested in CLIP (Radford et al., 2021a). The two evaluation metrics, Average and OOD Average, are computed by calculating the mean accuracy across all five datasets, as well as the four OOD datasets, excluding ImageNet.

| Method | ImageNet | ImageNet-A | ImageNet-V2 | ImageNet-R | ImageNet-S | Average | OOD Average |
|---|---|---|---|---|---|---|---|
| CLIP-ViT-B/16 | 66.73 | 47.87 | 60.86 | 73.98 | 46.09 | 59.11 | 57.20 |
| CLIP + E. | 68.77 | 51.05 | 62.21 | 77.54 | 48.25 | 61.56 | 59.76 |
| CoOp† | **71.51** | 49.71 | 64.20 | 75.21 | 47.99 | 61.72 | 59.28 |
| CoCoOp† | 71.02 | 50.63 | 64.07 | 76.18 | 48.75 | 62.13 | 59.91 |
| TPT | 68.48 | 54.49 | 63.26 | 75.51 | 47.90 | 61.93 | 60.29 |
| DiffTPT‡ | 70.30 | 55.68 | **65.10** | 75.00 | 46.80 | 62.28 | 60.52 |
| MTA | 69.32 | 57.20 | 63.60 | 76.88 | 48.53 | 63.11 | 61.55 |
| MTA + E. | 70.30 | **57.51** | 64.22 | 78.50 | 49.89 | 64.08 | 62.53 |
| GS-Bias | 69.02 | 54.55 | 63.37 | 76.64 | 48.21 | 62.36 | 60.69 |
| GS-Bias + E. | 70.57 | 56.61 | 64.62 | **80.49** | **50.33** | **64.52** | **63.01** |

*Table 3.* A comparison of the efficiency and effectiveness of our GS-Bias on ImageNet is shown. ∗ indicates results from reproducing the official released code, while ⋆ represents results from our optimizations, including prompt ensembling and text encoder reuse. All experiments are conducted on a single NVIDIA A800 GPU.

| Method | FPS(Image/Second) | Memory(Mib) | Accuracy(%) |
|---|---|---|---|
| TPT∗ | 1.38 | 19997 | 68.48 |
| MTA∗ | 8.59 | 3576 | 69.32 |
| MTA + E.⋆ | 12.55 | 1448 | 70.30 |
| GS-Bias + E.⋆ | 12.34 | 1308 | 70.57 |

*Table 4.* Ablation study of global bias $B_g$ and spatial bias $B_s$.

| $B_g$ | $B_s$ | ImageNet | 10 Cross-Datasets | Average |
|---|---|---|---|---|
| | | 68.77 | 65.64 | 67.21 |
| | ✓ | 69.10 | 65.77 | 67.44 |
| ✓ | | 70.45 | 66.07 | 68.26 |
| ✓ | ✓ | 70.57 | 67.03 | 68.80 |

supplementing bias knowledge while preserving the original CLIP output. Compared to the few-shot prompt tuning methods CoOp and CoCoOp, GS-Bias achieves substantial improvements of 3.15% and 2.40%, respectively, without requiring access to training data. Compared to the prompt optimizer TPT, GS-Bias demonstrates superior performance. When combined with prompt ensembling, it further exceeds the performance of DiffTPT, which incorporates a diffusion model. Furthermore, we achieve a 1.08% improvement over the visual optimizer MTA. These improvements verify that our GS-Bias can showcase stronger generalization on cross-dataset benchmarks.

**Results on the Domain Generalization.** We further assess the generalization capacity of GS-Bias across domains on ImageNet and its four variants. As shown in Table 2, GS-Bias demonstrates significant performance improvements across all scenarios. Compared to CoOp and CoCoOp trained on ImageNet, our method achieves improvements of 3.73% and 3.10% in OOD accuracy, respectively. When compared to other TTA methods, GS-Bias leads in both average and OOD accuracy, highlighting the robustness of our approach to domain shifts.

**Efficiency and Effectiveness.** We also provide a compari-

son of efficiency and effectiveness on ImageNet, evaluated on a single NVIDIA A800 GPU using the officially released code. As shown in Table 3, our GS-Bias benefits from lightweight backpropagation at the CLIP output logit level, achieving an FPS approximately 10 times faster than TPT, with only 6.5% of the memory overhead. Notably, compared to the parameter-free MTA, we achieve superior performance at nearly identical inference speeds.

### 4.3. Ablation Study

In this section, we provide a comprehensive analysis of the advantages of GS-Bias, examining the effectiveness of both global and spatial biases, and exploring the impact of different hyperparameter settings. All ablation studies are conducted across 11 benchmark datasets, including ImageNet and 10 datasets in cross-dataset benchmark. The experimental setup is consistent with those outlined in Section 4.1.

**The effects of Global Bias and Spatial Bias.** To verify the effectiveness of the proposed method, we examine the contributions of global bias and spatial bias. As shown in Table 4, the introduced global bias captures global semantics by aligning the augmented views for consistency, significantly enhancing CLIP's performance. The spatial bias, on the other hand, facilitates semantic exploration at the region level of the test image, further improving performance, particularly on unseen cross-dataset benchmarks. The com-

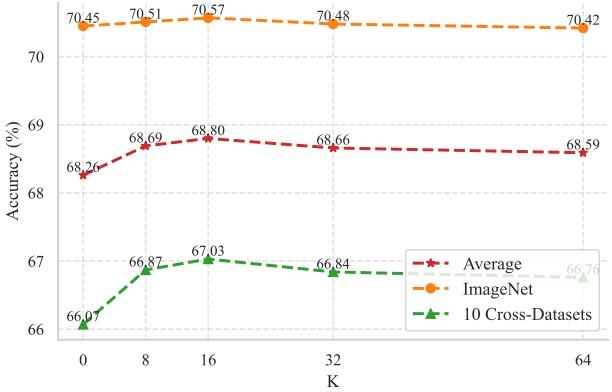

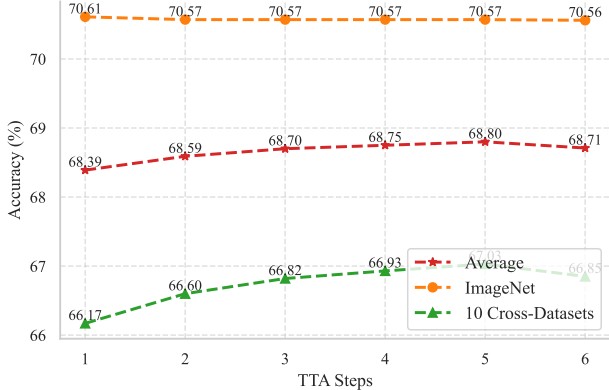

*Figure 3.* Ablation study on the hyperparameter $K$ in Eq. 12, where $K = 0$ represents the performance of $p_G(y|\boldsymbol{x})$ in Eq. 8.

*Table 5.* Ablation study on the hyperparameters $\alpha$ and $\beta$ in Eq.7 and Eq.14.

| $\alpha \ (\beta = 1)$ | 1 | 5 | 10 | 15 | 20 |
|---|---|---|---|---|---|
| ImageNet | 70.08 | 70.51 | 70.57 | 70.55 | 70.54 |
| 10 Cross-Datasets | 67.03 | 66.30 | 66.26 | 66.11 | 66.00 |
| Average | 68.56 | 68.41 | 68.42 | 68.33 | 68.27 |
| $\beta \ (\alpha = 1)$ | 0.1 | 0.5 | 1 | 1.5 | 2 |
| ImageNet | 70.04 | 70.08 | 70.12 | 70.17 | 69.97 |
| 10 Cross-Datasets | 66.15 | 66.48 | 67.03 | 66.97 | 66.67 |
| Average | 68.10 | 68.28 | 68.58 | 68.57 | 68.32 |

bination of both biases effectively enables high-performance TTA.

**The effects of $\alpha$ and $\beta$.** The test-time learning of GS-Bias is controlled by $\alpha$ and $\beta$, which represent the learning rates for global bias and spatial bias, respectively. To determine the effective configuration, we fixed one learning rate (*i.e.*, $\alpha = 1$ or $\beta = 1$) and observed how performance varied with changes in the other. As shown in Table 5, increasing the learning rate $\alpha$ leads to a continuous performance decline for GS-Bias on cross-dataset benchmarks, while on ImageNet, performance improves rapidly before stabilizing. This observation suggests that excessive capture of global information can lead to overfitting in unseen class scenarios, thereby impairing performance generalization. Conversely, learning more global semantics is crucial for general object classification. Furthermore, as learning rate $\beta$ increases, GS-Bias exhibits consistent performance improvements across both benchmarks before eventually declining. This pattern indicates that learning spatial semantics enriches visual concepts, but excessive focus on spatial semantics can disrupt the understanding of global semantics.

**The effects of $K$.** We investigate the effect of $K$ in eq. 12, which controls the scope of semantic consistency within the selected regions. As shown in Figure 3, recognition performance improves with increasing $K$, especially in cross-

*Figure 4.* Performance of GS-Bias at different TTA steps.

dataset generalization benchmarks, highlighting the beneficial role of spatial information in enhancing unseen class recognition. However, performance starts to decline when $K$ exceeds a certain threshold. This can be attributed to the fact that the foreground regions used for classification occupy only a portion of the test image, and selecting too many regions may introduce irrelevant background noise, which harms performance.

**The effects of TTA steps.** We also studied the impact of TTA steps on performance, where TTA steps refer to the number of backward propagation iterations. Unlike TPT, our method performs optimization solely at the CLIP output level, allowing for multi-step TTA with minimal overhead. In Figure 4, the accuracy on ImageNet remains stable after the first step of optimization. In contrast, on cross-datasets benchmarks, accuracy initially improves with increasing TTA steps, but begins to decline beyond a certain point, indicating that excessive learning steps may lead to overfitting during TTA.

## 5. Conclusion

In this work, we propose a novel Global-Spatial Bias Learner (GS-Bias) for test-time adaptation of vision-language models. Instead of modifying CLIP's textual prompts or visual features, our GS-Bias introduces two learnable biases directly at the output logits of CLIP. Particularly, the global bias captures the overall semantic features of a test image by aligning augmented views for consistency, while the spatial bias focuses on ensuring semantic coherence among regions within the spatial visual representation of the image. Moreover, our optimization operates solely at the CLIP output level, making the approach very lightweight. Extensive experiments demonstrate that GS-Bias consistently outperforms several state-of-the-art methods across all scenarios. We hope this study will inspire additional advancements in the field of multimodal learning and test-time adaptation.

## Acknowledgement

This work was supported by the National Science Fund for Distinguished Young Scholars (No.62025603), the National Natural Science Foundation of China (No. U21B2037, No. U22B2051, No. U23A20383, No. U21A20472, No. 62176222, No. 62176223, No. 62176226, No. 62072386, No. 62072387, No. 62072389, No. 62002305 and No. 62272401), and the Natural Science Foundation of Fujian Province of China (No. 2021J06003, No. 2022J06001).

## Impact Statement

This paper proposes GS-Bias, a Test-Time Adaptation (TTA) method for vision-language models that strikes a balance between performance and efficiency. The TTA community may benefit from the insights and findings presented in our research. There are many potential societal consequences of our work, none which we feel must be specifically highlighted here.

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

# A. Appendix

## A.1. More Quantitative Analysis

*Table 6.* Comparison of GS-Bias in Domain Generalization using ResNet50 (He et al., 2016) as the backbone. Bold values indicate the best performance, while † and ‡ refer to results reported in (Shu et al., 2022) and (Feng et al., 2023), respectively. E. denotes the use of hand-crafted prompts suggested in CLIP (Radford et al., 2021a). The two evaluation metrics, Average and OOD Average, are computed by calculating the mean accuracy across all five datasets, as well as the four OOD datasets, excluding ImageNet.

| Method | ImageNet | ImageNet-A | ImageNet-V2 | ImageNet-R | ImageNet-S | Average | OOD Average |
|---|---|---|---|---|---|---|---|
| CLIP-ResNet50 | 58.21 | 21.68 | 51.46 | 55.92 | 33.37 | 44.13 | 40.61 |
| CLIP + E. | 60.33 | 23.49 | 53.31 | 60.59 | 35.50 | 46.64 | 43.22 |
| CoOp† | **63.33** | 23.06 | 55.40 | 56.60 | 34.67 | 46.61 | 42.43 |
| CoCoOp† | 62.81 | 23.32 | 55.72 | 57.74 | 34.48 | 46.81 | 42.82 |
| TPT | 60.70 | 26.56 | 54.77 | 59.02 | 35.13 | 47.24 | 43.87 |
| DiffTPT‡ | 60.80 | **31.06** | 55.80 | 58.80 | **37.10** | 48.71 | 45.69 |
| MTA | 60.37 | 27.35 | 52.72 | 58.63 | 34.45 | 46.70 | 43.28 |
| MTA + E. | 61.44 | 28.02 | 54.88 | 61.88 | 36.37 | 48.52 | 45.29 |
| GS-Bias + E. | 62.05 | 27.83 | **55.91** | **63.01** | 36.98 | **49.16** | **45.93** |

In this section, we report our GS-Bias along with a quantitative analysis using ResNet50 (He et al., 2016) on domain generalization benchmark. To capture the spatial knowledge of ResNet50, we extract feature maps from the intermediate layer (with a size of 14×14×1024). The experimental setup follows the same configuration outlined in Section 4.1. The results in Table 6 show that the GS-Bias, when equipped with ResNet50, achieves state-of-the-art performance, further demonstrating the robustness of our method across different backbones.

## A.2. Qualitative results

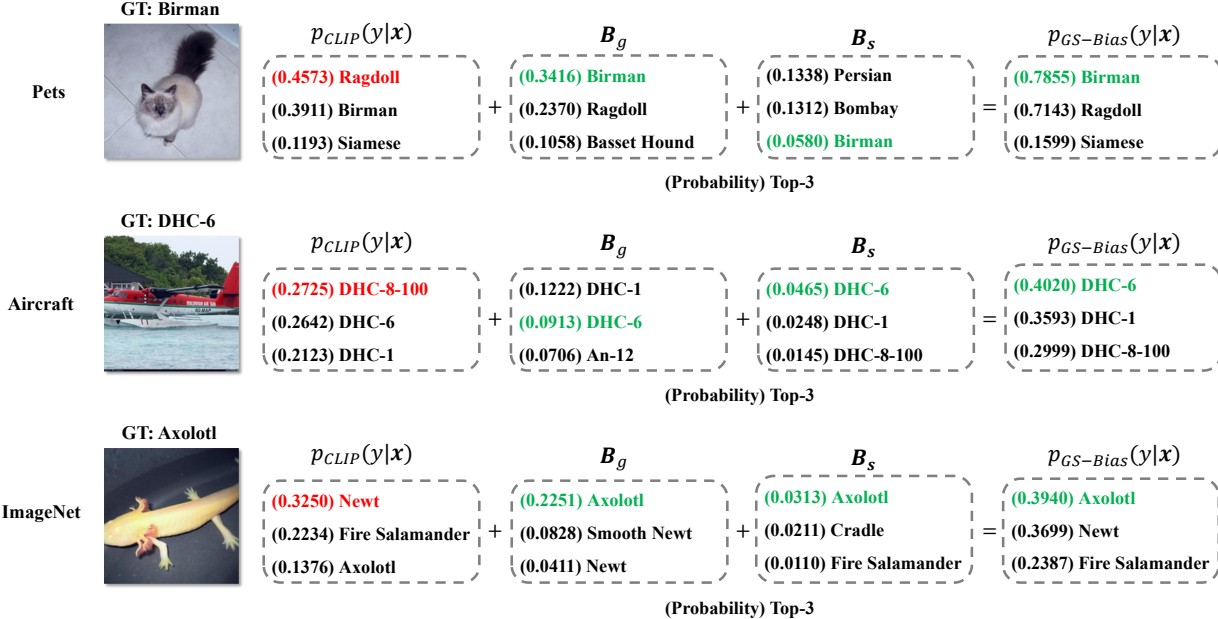

*Figure 5.* The inference examples of GS-Bias, which integrates the CLIP output $p_{\text{CLIP}}(y|\boldsymbol{x})$, global bias $\boldsymbol{B}_g$, and spatial bias $\boldsymbol{B}_s$.

We present several representative examples to demonstrate the effectiveness of GS-Bias during inference. As illustrated in Figure 5, distribution shifts between training and test data often lead CLIP to make biased predictions. In contrast, our method successfully rectifies such errors by incorporating both global bias and spatial bias. For instance, in a fine-grained

pet recognition task, CLIP misclassifies a Birman as a Ragdoll. However, the global bias component confidently guides the model toward the correct prediction by emphasizing the high-level semantic discrepancy, thereby correcting the mistake. Similarly, in an aircraft recognition scenario, where the key distinguishing features are subtle and localized (e.g., fine-grained text on the fuselage), CLIP tends to fail due to its insufficient sensitivity to spatial cues. Our spatial bias module captures such fine-grained visual concepts effectively, leading to accurate recognition even under challenging conditions.

## A.3. More Ablation Studies

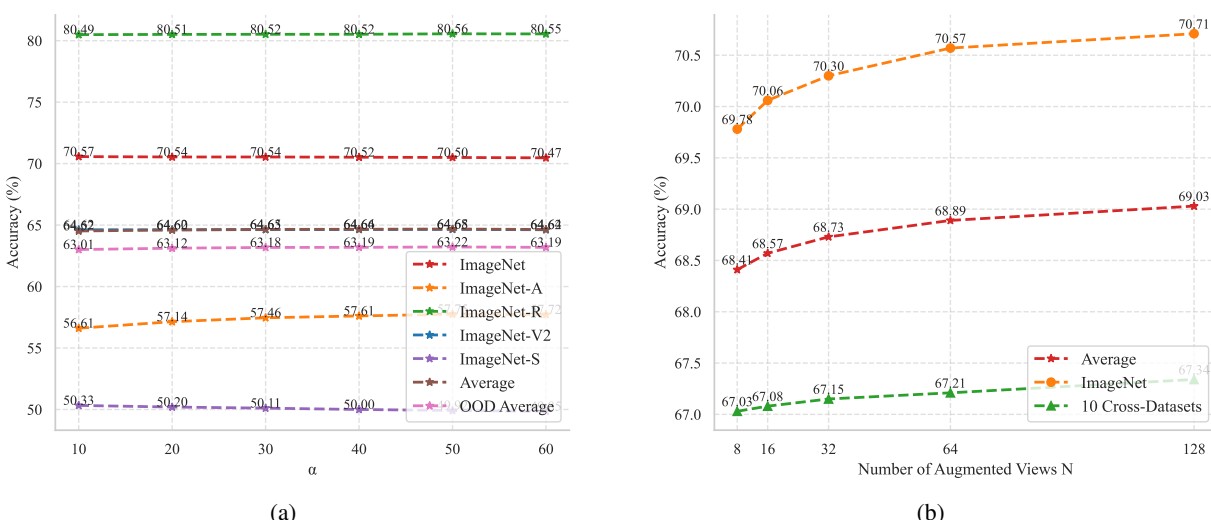

(a)                                                    (b)

*Figure 6.* More ablation studies. (a) Ablation study on the hyperparameter $\alpha$ in Eq. 7 for domain generalization. (b) Ablation study on the number of augmented views $N$ in Eq. 6.

**The effects of $\alpha$ on domain generalization.** In Figure 6 (a), we investigate the impact of $\alpha$ across multiple domain generalization datasets. We observe that as $\alpha$ increases, different datasets exhibit distinct trends. For example, the performance on ImageNet-A continuously improves, while that on ImageNet-S steadily declines. This suggests that the degree of global semantic learning varies across datasets. Adversarial datasets, for instance, require more extensive learning, while simpler datasets, like sketches, demand less.

**The effects of $N$.** Furthermore, we investigate how the number of augmented views affects performance. Intuitively, increasing the number of views enhances diversity and improves the overall representational quality. As shown in Figure 6 (b), the performance of GS-Bias improves consistently on both cross-dataset generalization and ImageNet as $N$ increases. The improvement on ImageNet is especially notable. These results suggest that GS-Bias can generalize well to unseen classes with only a few views. However, a larger number of diverse views is crucial for recognizing the broader range of classes in ImageNet.

*Table 7.* The number of significant spatial regions across 11 datasets.

|  | Flowers102 | DTD | Pets | Cars | UCF101 | Caltech101 | Food101 | SUN397 | Aircraft | EuroSAT | ImageNet | Average |
|---|---|---|---|---|---|---|---|---|---|---|---|---|
| $\tilde{K}_a$ | 18.03 | 12.83 | 18.43 | 16.42 | 16.23 | 16.48 | 16.80 | 18.56 | 15.95 | 11.91 | 19.12 | **16.43** |

**Analysis of spatial bias.** In Table 7, we analyze the number of spatial regions that are significantly associated with the classification target across different datasets. Specifically, we normalize the similarity scores $M$ in Eq. 10 and define regions with scores greater than 0.1 as significant, denoted by $\tilde{K} = \sum_i \mathbb{I}\left(\frac{M_i - \min(M)}{\max(M) - \min(M)} > 0.1\right)$. Furthermore, we compute the average number of significant regions for each dataset, represented as $\tilde{K}_a$. The results indicate that the number of significant regions varies across different datasets. For example, the low-resolution satellite images in EuroSAT appear blurry, making spatial information less distinguishable and resulting in fewer significant regions. In contrast, fine-grained datasets such as pet and flower recognition yield a greater number of significant regions. Furthermore, we observe that the average number of significant regions across the 11 datasets is approximately 16. Notably, this setting also yields the best performance in Figure 3, making it a reasonable and well-justified choice in our experiments.

*Table 8.* Comparison between our spatial views and small-scale crop-based augmented views.

| Method | 10 Cross-Datasets | ImageNet |
|---|---|---|
| $Crop_{1\times1}$ | 66.02 | 70.49 |
| $Crop_{2\times2}$ | 66.03 | 70.48 |
| GS-Bias (w/o $B_s$) | 66.07 | 70.45 |
| GS-Bias | 67.03 | 70.57 |

*Table 9.* Quantitative analysis of the method equipped with historical data stream.

| Method | Memory | 10 Cross-Datasets | ImageNet |
|---|---|---|---|
| TDA | 1058M | 67.96 | 69.50 |
| DPE | 6560M | 68.13 | 71.08 |
| GS-Bias | 1308M | 67.03 | 70.57 |
| GS-Bias + TDA | 1308M | 68.30 | 70.96 |

Furthermore, we analyze the differences between spatial views and small-scale crop-based augmented views. To this end, we generate $1 \times 1$ and $2 \times 2$ scale augmented views via random cropping and use them as substitutes for spatial views in the learning of spatial bias. As shown in Table 8, our spatial bias effectively improves performance compared to GS-Bias without spatial bias, especially in unseen cross-dataset generalization (66.07% vs. 67.03%), demonstrating its ability to capture visual concepts missed by global bias. In contrast, simply reducing crop size does not achieve the same effect. This is because we learn spatial bias from the spatial features of the vision encoder, where each region inherently encodes rich contextual information. In comparison, image-level cropping produces isolated patches that lack spatial coherence. Moreover, our region selection is guided by the correlation between spatial regions and class descriptions, whereas cropping is purely random.

**Compatibility of GS-Bias with Historical Data Streams.** Recent VLM-based TTA methods have leveraged historical data streams to achieve stronger performance. For instance, TDA (Karmanov et al., 2024) utilizes historical information to update a dynamic queue for training-free TTA, while DPE (Zhang et al., 2024a) and HisTPT (Zhang et al., 2024b) further extend this paradigm by incorporating prototype learning and prompt tuning, respectively. In contrast, our GS-Bias operates solely on individual test samples without accessing historical data, making it more aligned with methods such as TPT (Shu et al., 2022) and MTA (Zanella & Ben Ayed, 2024). Notably, GS-Bias remains compatible with historical data streams, allowing for potential integration. In Table 9, we report the performance of GS-Bias, TDA, and DPE across 11 datasets, along with their memory overhead on ImageNet, where all methods adopt the same text prompts for fair comparison. We also present a coarse combination of TDA and GS-Bias to simulate the effect of incorporating historical data. The results show that our method achieves a favorable balance between performance and efficiency. Moreover, when equipped with historical data, GS-Bias achieves further improvements without increasing memory consumption. This is because our bias learning is performed solely at the output level, effectively avoiding the need to store gradient-bearing objects in the queue.

## A.4. Limitation

We further discuss the unexplored limitations of our proposed GS-Bias, which will be our future focus. Although GS-Bias achieves a favorable trade-off between performance and efficiency, one notable limitation is its reliance on empirically selected hyperparameters. Despite extensive ablation studies across diverse datasets indicating that GS-Bias is relatively insensitive to these settings, hyperparameters still need to be reconfigured for different test samples. While this empirical strategy offers a practical baseline, it may not deliver optimal performance for every individual instance. Future work could explore dynamic, sample-specific hyperparameter adjustment mechanisms to further enhance the model's adaptability and generalization.

