# OpenReview forum: "GS-Bias: Global-Spatial Bias Learner for Single-Image Test-Time Adaptation of Vision-Language Models"
_ICML.cc/2025/Conference — ICML 2025 poster_

### Official Review · Reviewer_pUFB · 2025-03-10

**Overall Recommendation:** 3

**Summary:**

This paper introduces Global-Spatial Bias Learner (GS-Bias), a test-time adaptation (TTA) method designed to improve the zero-shot generalization of vision-language models (VLMs) like CLIP, while keeping computational costs low. The core innovation is the addition of two learnable biases—global bias and spatial bias—to the model’s output logits during testing, without the need of training data. Global Bias captures broad semantic patterns by aligning predictions across multiple augmented views of a test image, optimized via entropy minimization of high-confidence logits. Spatial Bias enhances local understanding by focusing on task-relevant regions within the image’s spatial features, ensuring the regional consistency. Both biases are applied directly to the pre-trained VLM’s logits, avoiding full-network backpropagation and making the method highly efficient.GS-Bias outperforms state-of-the-art TTA methods like TPT and MTA across 15 benchmark datasets, boosting cross-dataset generalization by 2.23% over TPT and domain generalization by 2.72%, while using only 6.5% of TPT’s memory on ImageNet. It excels in zero-shot and domain-shift scenarios, balancing performance and efficiency effectively. The method’s low memory footprint and fast inference speed make it practical for real-world use. In essence, GS-Bias offers a lightweight, powerful TTA solution that significantly enhances VLM generalization with minimal computational overhead.

**Claims And Evidence:**

Supported Claims
State-of-the-Art Performance:
Evidence: Tables 1 and 2 show GS-Bias outperforming TPT, MTA, and training-time methods (CoOp, CoCoOp) across 15 benchmarks, with 67.03% average accuracy on cross-datasets and 68.80% on domain generalization.
Efficiency Advantages:
Evidence: Table 3 reports 12.34 FPS and 1,308 MiB memory usage, significantly better than TPT (1.38 FPS, 19,997 MiB).
Problematic Claims or Gaps
1. The concept of global prediction consistency, which underpins the global bias mechanism discussed in the contributions section, has already been extensively adopted in the TTA (Test-Time Adaptation) domain. Positioning this as a novel contribution lacks sufficient innovation, as it represents a well-established methodology rather than a substantive advancement in the field.
2. The local bias mechanism proposed in the contributions section appears fundamentally equivalent to enforcing predictive consistency constraints at finer spatial scales. This raises a critical methodological question: Would comparable effects be achieved simply by reducing crop sizes in spatial bias augmentation operations, rather than implementing the described local bias paradigm? Given this equivalence, the claimed innovation of local bias demonstrates incremental advancement rather than substantiative methodological distinction.

**Essential References Not Discussed:**

There are indeed a few related works that, while not discussed in the paper, provide important context for its key contributions. For instance, although the paper emphasizes output-level adaptation via learnable biases, methods such as TENT (Wang et al., 2021) show that adapting only batch normalization parameters through entropy minimization can effectively counter domain shifts with minimal network changes. Furthermore, several recent VLM TTA methods that emphasize both accuracy and efficiency, such as DPE, TDA, and HisTPT, have not been discussed. These approaches exhibit significant improvements in performance and efficiency compared to earlier methods like TPT and DiffTPT, warranting more comprehensive discussion and comparative analysis.

**Experimental Designs Or Analyses:**

Baseline Comparisons:

Strengths:
Includes established methods (TPT, MTA, CoOp, CoCoOp) for fair comparison.

Weaknesses:
1. The manuscript lacks comparisons with the state-of-the-art VLM TTA methods, such as DPE, TDA, and HisTPT.
2. In Tables 1 and 2, the performance of the original GS-Bias method does not demonstrate substantial improvements over TPT and MTA; only after applying an ensemble with a hand-crafted template is a significant enhancement observed, which raises concerns about the intrinsic effectiveness of the proposed method.

[1] DPE: Dual Prototype Evolving for Test-Time Generalization of Vision-Language Models. NIPS 2024
[2] TDA: Efficient Test-Time Adaptation of Vision-Language Models. CVPR 2024
[3] HisTPT: Historical Test-time Prompt Tuning for Vision Foundation Models. NIPS 2024

**Methods And Evaluation Criteria:**

The proposed method (GS-Bias) and its evaluation criteria are well-aligned with the problem and application of test-time adaptation (TTA) for vision-language models (VLMs). Here’s a breakdown of their rationale and suitability:
Problem-Specific Design:
Core Issue Addressed: Existing TTA methods struggle to balance performance (e.g., cross-dataset generalization) and efficiency (e.g., memory usage, inference speed).
Efficiency vs. Performance Trade-Off:
By design, GS-Bias reduces memory usage to 6.5% of TPT and achieves 10× faster inference (12.34 FPS vs. 1.38 FPS), addressing the inefficiency of prompt optimizers and instability of visual optimizers.
Evaluation Criteria and Benchmark Suitability
The paper evaluates GS-Bias on 15 datasets, covering two critical scenarios:
Cross-Dataset Generalization and Domain Generalization:Tests generalization to unseen classes and tasks, reflecting real-world scenarios where VLMs encounter novel distributions.Validates robustness to distribution shifts (e.g., adversarial corruptions, sketch-like inputs), critical for real-world deployment under dynamic conditions.
Efficiency Metrics:
Memory Usage and FPS are explicitly measured, aligning with practical constraints for edge or real-time applications.

**Other Comments Or Suggestions:**

No

**Other Strengths And Weaknesses:**

Strengths:
1. The paper is well-organized and is easy to read. Evaluation includes 2 set of experiments on 15 dataset which is comprehensive.
2. Efficiency through Logit-Level Adaptation: The paper contributes a significant efficiency gain by restricting the adaptation to the logit outputs of the pre-trained model (CLIP). This approach is inspired by the desire to minimize computational overhead while still leveraging the benefits of test-time adaptation.

Weaknesses：
1. The proposed global bias mechanism, which relies on global prediction consistency, is not a novel idea since it is already widely used in the TTA domain.
2. The local bias mechanism seems equivalent to enforcing consistency at a finer spatial scale, raising the question of whether simply reducing crop sizes would yield similar effects.
3. The manuscript lacks comparisons with the state-of-the-art VLM TTA methods, such as DPE, TDA, and HisTPT.
4. In Tables 1 and 2, the performance of the original GS-Bias method does not demonstrate substantial improvements over TPT and MTA; only after applying an ensemble with a hand-crafted template is a significant enhancement observed, which raises concerns about the intrinsic effectiveness of the proposed method.

**Questions For Authors:**

1. In Table 4, the ablation study indicates that spatial bias yields only about a 0.2% improvement on average. Does this suggest that the positive effect of spatial bias is minimal, or that it merely acts as a smaller-scale version of global bias? The authors should include additional experiments to substantiate the method's effectiveness.
2. In Table 1, GS-Bias is evaluated with a batch size of 8, while TPT and MTA experiments use a batch size of 64. The absence of BS=64/BS=128 results in the ablation study is concerning; the authors should provide further results or justification to address this discrepancy.

**Relation To Broader Scientific Literature:**

The paper's key contributions lie in introducing lightweight, learnable global and spatial biases that act directly on the CLIP model's output logits to achieve test-time adaptation without the heavy full-network back-propagation or complex visual feature optimization seen in methods like TPT, DiffTPT, and MTA. The global bias leverages the idea of multi-view semantic consistency to enhancing textual prompts and strengthen overall semantic representation, while the spatial bias utilizes local region information from the visual encoder to focus on target classes, thus boosting generalization across domains and unseen classes. This approach builds on and extends earlier research in zero-shot and cross-domain adaptation, improving both efficiency and robustness.

**Theoretical Claims:**

While the study demonstrates that the merits in its application-oriented focus on Vision-Language Models (VLMs), this prioritization has resulted in a notable absence of formal theoretical propositions and their systematic substantiation. The methodological framework would significantly benefit from rigorous theoretical grounding to complement its empirical implementation, as current arguments remain predominantly heuristic rather than axiomatically derived

---

> ### Author Rebuttal · Authors · 2025-03-31
>
> We greatly appreciate the reviewer’s encouraging and valuable comments on our paper. Below, we address the raised concerns.
>
> **Q1: The proposed global bias mechanism, which relies on global prediction consistency, is not a novel idea.**
>
> **A1:** We acknowledge that relying on global prediction consistency is not novel. However, our key innovation lies in introducing a **learnable global bias at the output level, which effectively addresses the efficiency bottleneck in prior methods**. Previous approaches require **modifying model inputs**, leading to substantial computational overhead. In contrast, our method operates **purely at the output level**, eliminating the need for expensive backpropagation while still ensuring effective test-time adaptation. Thus, our method strikes an effective balance between performance and efficiency. For instance, it improves cross-dataset and domain generalization by 2.23% and 2.72% over TPT, while using only 6.5% of its memory on ImageNet.
>
> **Q2: Could simply reducing the crop size yield a similar effect on spatial bias?**
>
> **A2:** Thank you for your question. In response, we conducted experiments on 11 datasets by reducing the crop size to $1\times1$ and $2\times2$ for spatial bias learning. The results show that our spatial bias effectively improves performance compared to GS-Bias without spatial bias, especially in unseen cross-dataset generalization (66.07% vs. 67.03%), demonstrating its ability to capture visual concepts missed by global bias. In contrast, simply reducing crop size does not achieve the same effect.
>
> This is because we learn spatial bias from the spatial features of the vision encoder, where each region inherently encodes rich contextual information. In comparison, image-level cropping produces isolated patches that lack spatial coherence. Moreover, our region selection is guided by the correlation between spatial regions and class descriptions, whereas cropping is purely random.
> |Method|10 Cross-Datasets|ImageNet|
> |-|-|-|
> |$Crop_{1\times1}$|66.02|70.49|
> |$Crop_{2\times2}$|66.03|70.48|
> |GS-Bias (w/o $B_s$)|66.07|70.45|
> |GS-Bias|**67.03**|**70.57**|
>
> **Q3: Missed related works.**
>
> **A3:** We have conducted a comparison with state-of-the-art VLM TTA methods, including DPE, TDA, and HisTPT. As summarized below, TDA utilizes historical data streams to update a dynamic queue for training-free TTA, while DPE and HisTPT extend this by incorporating prototype learning and prompt tuning, respectively. In contrast, GS-Bias operates purely on a single image without accessing historical data, making it more aligned with TPT and MTA. Notably, GS-Bias remains compatible with TDA, allowing for potential integration.
>
> To further clarify our advantages, we report the performance and memory cost of GS-Bias with TDA and DPE on ImageNet, and combine TDA and GS-Bias as a rough version with historical data flow. The results show that GS-Bias strikes a strong balance between performance and efficiency. Furthermore, when equipped with historical data flow, GS-Bias improves further without increasing memory usage. This is due to our bias learning operates solely at the output level, effectively avoiding the storage of gradient-bearing objects in the queue.
> |Method|Memory|Accuracy|
> |-|-|-|
> |TDA|1058M|69.5|
> |DPE|6560M|71.2|
> |GS-Bias|1308M|70.6|
> |GS-Bias-History|1308M|71.0|
>
> **Q4:  The performance of the original GS-Bias does not show substantial improvements over TPT and MTA.**
>
> **A4:** As shown in the table below, we aggregated the performance of the original GS-Bias across 15 datasets (Fig 1, Tab 1, and Tab 2 in the manuscript). With the same settings (BS=64), our method consistently outperforms others. More importantly, **GS-Bias requires only 6.5% of TPT’s memory on ImageNet and achieving a $10\times$ speedup**, demonstrating much higher efficiency. Overall, even in its original form, GS-Bias achieves an substantial improvement for the trade-off between performance and efficiency compared to TPT and MTA.
> |BS|Accuracy|
> |-|-|
> |TPT|63.84|
> |MTA|63.99|
> |GS-Bias|**64.23**|
>
> **Q5: The positive effect of spatial bias is minimal?**
>
> **A5:** We elaborate on the effectiveness of spatial bias in **A2**. Regarding concerns about the performance gain, we clarify that spatial bias is designed to complement global bias, not to be used independently in CLIP. Since CLIP is trained on global features, its spatial distributions are smoother, leading to weaker gradient updates. Thus, applying spatial bias directly to CLIP has limited benefits.
>
> **Q6: More ablation studies on BS.**
>
> **A6:** We list all the results for BS=8,16,32,64, and 128 below. It turns out that a larger BS leads to better performance. GS-Bias outperforms both TPT and MTA even in the simplest setup.
> |BS|8|16|32|64|128|
> |-|-|-|-|-|-|
> |GS-Bias|64.86|64.93|65.26|65.16|65.38|
> |GS-Bias + E.|67.03|67.08|67.15|67.21|67.34|
>
> We will include the added results and discussions in the final version as supplementary material.

---

### Official Review · Reviewer_HBpX · 2025-03-14

**Overall Recommendation:** 3

**Summary:**

This paper introduces Global-Spatial Bias Learner (GS-Bias), a test-time adaptation method for vision-language models (VLMs). GS-Bias's main idea is to learn two biases at the output logits of CLIP:
- Global bias that captures semantic consistency across augmented views of a test image
- Spatial bias that learns semantic coherence between regions in the image's spatial representation

Experiment results show that compared to previous methods (TPT, MTA), GS-Bias is more memory efficient and achieves better performance in general.

**Claims And Evidence:**

The paper's claims about efficiency and performance improvements are well-supported by experiments.

1. Efficiency claims: Figure 1 (b) and Table 3
2. Performance improvements claims: Table 1 and Table 2.
3. Besides, ablation studies effectively validate the contribution of both global and spatial biases: Table 4.

**Essential References Not Discussed:**

N/A

**Experimental Designs Or Analyses:**

The experimental design appears sound with appropriate controls and comparisons, validating the efficiency and performance improvements.

**Methods And Evaluation Criteria:**

The methods and evaluation criteria are appropriate for the task, as they are commonly used in the field.

**Other Comments Or Suggestions:**

N/A

**Other Strengths And Weaknesses:**

Weaknesses:
1. There might be a typo (GB-Bias) in the title of the paper.
2. The paper could benefit from showing concrete inference examples to illustrate the method's effectiveness, rather than purely relying on numbers.
3. The paper has limited discussion of potential failure cases or limitations.

**Questions For Authors:**

N/A

**Relation To Broader Scientific Literature:**

I believe this work can benefit the field of VLMs as it provides a new approach for efficient test-time adaptation.

**Theoretical Claims:**

N/A

---

> ### Author Rebuttal · Authors · 2025-03-29
>
> **Q1: A typo in the title of the paper.**
>
> **A1:** We sincerely appreciate your thorough and responsible review of our manuscript. We apologize for the typo caused by our oversight, and we will correct "GB-Bias" to "GS-Bias" in the final version.
>
> **Q2: Provide concrete inference examples to illustrate the effectiveness of the proposed method.**
>
> **A2:** Thank you very much for your insightful suggestions, which have encouraged us to present our work more intuitively. Following your advice, we have provided seven concrete inference examples, with images sourced from seven different datasets (ImageNet, ImageNet-A, ImageNet-S, Pets, Aircraft, Flowers102, and EuroSAT). For your convenience, we have placed these examples in the following anonymous link: [Inference Examples](https://github.com/anonymoussubmission74/Inference-Examples/blob/main/Inference_Examples.png).
>
> We emphasize that GS-Bias consists of three key components: CLIP output, global bias, and spatial bias. To intuitively demonstrate the effectiveness of bias learning, we present the Top-3 probabilities and their corresponding categories for each component. The examples show that the combination of global and spatial biases effectively corrects the erroneous outputs of the original CLIP model.
>
> **Q3: The paper has limited discussion of potential failure cases or limitations.**
>
> **A3:** Thank you for your constructive comments, which has encouraged us to provide a more comprehensive discussion of GS-Bias. While GS-Bias achieves a well-balanced trade-off between performance and efficiency, one notable limitation is that the selection of hyperparameters is based on empirical choices. Although we performed ablation studies on the hyperparameters using as many datasets as possible and found that the model's performance is not sensitive to the hyperparameters, it is still necessary to reselect the hyperparameters for different test samples.
>
> For instance, we set the learning rates of global and spatial biases to a fixed value of $ \alpha = 1 $ and $ \beta = 1 $ to achieve cross-dataset generalization. However, some samples may favor learning more from global information, while others may require a stronger focus on spatial information.  An empirically fixed setting might lead to suboptimal adjustments. To illustrate this more intuitively, we provide two failure cases (link: [Failure Cases](https://github.com/anonymoussubmission74/Inference-Examples/blob/main/failure_cases.png)), where fine-grained aircraft recognition tends to rely more on spatial information, whereas action recognition benefits more from global information.
>
> Thus, we acknowledge that such empirical selection may not be optimal for every individual data sample, but it serves as a practical starting point. Future research could explore dynamic strategies for adjusting the balancing hyperparameters on a per-sample basis to further enhance model performance.
>
> Once again, we sincerely appreciate your professional review. We will incorporate the discussion on potential failure cases and limitations into the final version and provide more concrete inference examples.

---

### Official Review · Reviewer_F3JS · 2025-03-16

**Overall Recommendation:** 4

**Summary:**

This paper introduces GS-Bias, a novel test-time adaptation (TTA) method for Vision-Language Models (VLMs). The approach aims to improve zero-shot generalization by learning two biases: a global bias that captures the global semantic features of a test image through consistency across augmented views, and a spatial bias that learns semantic coherence between regions in the image's spatial representation. GS-Bias adds these biases directly to the logits output of the pre-trained VLM, avoiding computationally expensive full backpropagation. The authors claim that GS-Bias achieves state-of-the-art performance on several benchmark datasets while being highly efficient in terms of memory usage.

**Claims And Evidence:**

The claims made in the paper are generally supported by the evidence provided.

The performance improvements over other TTA methods, as reported in Tables 1 and 2, seem consistent and significant, supporting the effectiveness claim.

The efficiency claim is supported by the memory usage comparison in Figure 1 (b) and the FPS comparison in Table 3.

The ablation studies in Table 4 and Figure 3, demonstrating the contributions of both global and spatial biases, are also convincing.

However, the number of selected spatial region is limited. The authors could have more comprehensive analysis on the number of spatial region.

**Essential References Not Discussed:**

No

**Experimental Designs Or Analyses:**

The experimental design appears sound.

The authors compare GS-Bias with several strong baselines, including zero-shot CLIP, training-time adaptation methods, and other TTA methods.

Ablation studies are conducted to analyze the contributions of different components of GS-Bias (global bias, spatial bias, hyperparameters).

The experiments cover a range of datasets and tasks, providing a comprehensive evaluation of the method's generalization ability.

A direct comparison of efficiency in terms of total computational time is missing. Including this would enhance the practicality insight of GS-Bias.

**Methods And Evaluation Criteria:**

The proposed method, GS-Bias, is well-motivated. The idea of learning biases directly at the logit level is an efficient way to adapt VLMs during test time. The combination of global and spatial biases seems reasonable for capturing both overall semantics and local details.

The evaluation criteria are standard for this task. The paper uses several established benchmark datasets for cross-dataset generalization and domain generalization. Top-1 accuracy is a common metric for classification performance.

**Other Comments Or Suggestions:**

It would be interesting to see how GS-Bias performs on more complex and fine-grained tasks, such as object detection or semantic segmentation.

**Other Strengths And Weaknesses:**

Strengths:

-The idea of learning biases at the logit level for test-time adaptation is novel and efficient.

-The paper addresses an important problem (zero-shot generalization of VLMs) and proposes a practical solution that achieves state-of-the-art performance.

-The paper is well-written and easy to understand, with clear explanations of the method and experimental results.

Weaknesses:

-The hyperparameter selections lack comprehensive analysis on the spatial region.

**Questions For Authors:**

In the analysis of the selected regions, can you provide more reasons on the setting of the number of regions?

**Relation To Broader Scientific Literature:**

The paper does a good job of situating GS-Bias within the broader scientific literature.

The authors discuss the related works in prompt tuning, adapter-based methods, and test-time adaptation for VLMs.

They clearly explain how GS-Bias differs from and improves upon existing TTA methods by addressing the limitations of prompt tuning and visual optimization approaches.

The paper also cites relevant works on contrastive visual-language models and related techniques (e.g., MeanShift).

**Theoretical Claims:**

There are no theoretical claims.

---

> ### Author Rebuttal · Authors · 2025-03-28
>
> **Q1: Provide a more comprehensive analysis of the number of spatial regions and the reasons for setting of the number of regions.**
>
> **A1:** Thank you for your insightful comments. In response, we have expanded our analysis by incorporating two new experiments:
> - **Exp1:** Computing the number of spatial regions that are significantly related to the classification target across 11 datasets.
> - **Exp2:** Extending Figure 3 by adding results for $K$ = 4, 128, and 196. (ViT-B/16 has 196 spatial regions)
>
> Following Eq.10 and 11 in the manuscript, we select the top-K regions for spatial bias learning by computing the similarity scores $\boldsymbol{M}$ between all spatial regions and the class descriptions. Higher scores indicate stronger relevance to the classification target, while lower scores may correspond to irrelevant regions.
>
> In **Exp1**, we normalize the similarity scores and consider regions with scores greater than 0.1 as significant, denoted by $\tilde{K} = \sum_{i} \mathbb{1} \left( \frac{\boldsymbol{M}_i - \min(\boldsymbol{M})}{\max(\boldsymbol{M}) - \min(\boldsymbol{M})} > 0.1 \right)$. Furthermore, we compute the average number of significant regions for each dataset, represented as $\tilde{K}_a$. The results indicate that the number of significant regions varies across different datasets. For example, the low-resolution satellite images in EuroSAT appear blurry, making spatial information less distinguishable and resulting in fewer significant regions. In contrast, fine-grained datasets such as pet and flower recognition provide a greater number of significant regions. Furthermore, we observe that the average number of significant regions across the 11 datasets is approximately 16.
>
> **Exp2** further confirms that as $K$ increases, performance initially improves, reaches a peak, and then starts to decline. This suggests that incorporating an appropriate number of spatial regions provides beneficial class-related information, whereas an excessively large $K$ (e.g., $K=196$) may lead to severe overfitting, causing the optimization to be trapped in irrelevant, misleading information.
>
> In summary, we observe that $K$ = 16 achieved the best average performance and exhibited significant relevance, making it a reasonable and well-justified choice. We list the results as below.
>
> - **Exp1. The number of significant spatial regions $\tilde{K}_a$ across 11 datasets.**
> |Method|Flower102|DTD|Pets|Cars|UCF101|Caltech101|Food101|SUN397|Aircraft|EuroSAT|ImageNet|Average|
> |-|-|-|-|-|-|-|-|-|-|-|-|-|
> |$\tilde{K}_a$|18.03|12.83|18.43|16.42|16.23|16.48|16.80|18.56|15.95|11.91|19.12|**16.43**|
>
> - **Exp2. Results of different numbers of spatial regions $K$ across 11 datasets.**
> |$K$|0|4|8|16|32|64|128|196|
> |-|-|-|-|-|-|-|-|-|
> |10 Cross-Datasets|66.07|66.84|66.87|**67.03**|66.84|66.76|66.75|66.70|
> |ImageNet|70.45|70.52|70.51|**70.57**|70.48|70.42|70.38|70.35|
> |Average|68.26|68.68|68.69|**68.80**|68.66|68.59|68.57|68.53|
>
> **Q2: Comparison of efficiency in total computational time.**
>
> **A2:** Thank you for your valuable suggestions. To further demonstrate the practicality of our method, we have supplemented our analysis by reporting the total computation time of GS-Bias, MTA, and TPT across 11 datasets. Specifically, we set the augmentation batch size to 8 for cross-dataset generalization and 64 for ImageNet. All experiments were conducted on a single RTX 3090 GPU. The results indicate that GS-Bias achieves a significant speedup compared to TPT, while its total computational cost remains nearly identical to that of the parameter-free MTA. The detailed results are presented below.
>
> - **Comparison of efficiency in total computational time.**
> |Method|Flower102|DTD|Pets|Cars|UCF101|Caltech101|Food101|SUN397|Aircraft|EuroSAT|ImageNet|
> |-|-|-|-|-|-|-|-|-|-|-|-|
> |TPT|4min|2min|5min|23min|6min|4min|60min|103min|6min|10min|660min|
> |MTA|1min|1min|1min|3min|1min|1min|12min|9min|1min|3min|85min|
> |GS-Bias|1min|1min|2min|3min|2min|1min|12min|10min|2min|4min|88min|
>
> **Q3: How GS-Bias performs on more complex and fine-grained tasks？**
>
> **A3:** Thank you for your insightful suggestion. The idea of GS-Bias can be extended to other foundation models for various downstream tasks. For example, in segmentation tasks, a learnable bias $B \in R^{1 \times W \times H \times C}$ can be incorporated into the output segmentation mask $M \in R^{1 \times W \times H \times C}$, making it an updatable mask $\tilde{M} \in R^{1 \times W \times H \times C}$. Applying a segmentation-specific test-time objective to $\tilde{M}$ facilitates efficient bias learning.
>
> However, applying GS-Bias to more complex and fine-grained tasks requires designing a new test-time objective that aligns with the nature of the model and the specific downstream task. We plan to explore this direction further in future research.
>
> We sincerely appreciate your profound comments and will incorporate the above discussion into the final version of the paper.

---

> > ### Comment · Reviewer_F3JS · 2025-04-09
> >
> > Thanks for providing the detailed rebuttal. My concerns have been fully addressed. Therefore, I would increase my rating to Accept (4).

---

### Decision · Program_Chairs · 2025-05-01

**Decision:**

Accept (poster)

**Comment:**

This paper proposes GS-Bias, a novel test-time adaptation (TTA) method for Vision-Language Models (VLMs). The core idea of GS-Bias is to learn two types of biases at the output logits of CLIP: a global bias, which captures semantic consistency across augmented views of a test image, and a spatial bias, which models semantic coherence between regions in the image's spatial representation. Experimental results demonstrate that GS-Bias is memory-efficient and delivers strong performance.
Following the rebuttal, the paper received all positive recommendations (one Accept and two Weak Accepts). Concerns related to reduced crop sizes and the intrinsic effectiveness of the proposed method were adequately addressed by the authors. The Area Chair concurs with the reviewers' assessments and recommends accepting the paper.